# Validation of an Analytical Method for Nitrite and Nitrate Determination in Meat Foods for Infants by Ion Chromatography with Conductivity Detection

**DOI:** 10.3390/foods9091238

**Published:** 2020-09-04

**Authors:** Donatella Coviello, Raffaella Pascale, Rosanna Ciriello, Anna Maria Salvi, Antonio Guerrieri, Michela Contursi, Laura Scrano, Sabino A. Bufo, Tommaso R.I. Cataldi, Giuliana Bianco

**Affiliations:** 1Department of Science, University of Basilicata, Via dell’Ateneo Lucano, 10, 85100 Potenza, Italy; donatella.coviello@unibas.it (D.C.); rosanna.ciriello@unibas.it (R.C.); anna.salvi@unibas.it (A.M.S.); antonio.guerrieri@unibas.it (A.G.); michela.contursi@unibas.it (M.C.); sabino.bufo@unibas.it (S.A.B.); giuliana.bianco@unibas.it (G.B.); 2Department of European and Mediterranean Cultures: Architecture, Environment, Cultural Heritage, University of Basilicata, Via Lanera, 20, 75100 Matera, Italy; laura.scrano@unibas.it; 3Department of Geography, Environmental Management and Energy Studies, University of Johannesburg, P.O. Box 524, Auckland Park 2006, Johannesburg 2028, South Africa; 4Department of Chemistry, University of Bari, Via Orabona, 4, 70126 Bari, Italy; tommaso.cataldi@uniba.it

**Keywords:** Ion chromatography, conductivity detection, homogenized meat, baby food, nitrite, nitrate

## Abstract

Nitrate and nitrite as sodium or potassium salts are usually added to meat products to develop the characteristic flavor, to inhibit the growth of microorganisms (particularly *Clostridium botulinum*), and effectively control rancidity by inhibiting lipid oxidation. However, both nitrate and nitrite ions need to be monitored for ensuring the quality and safety of cured meats. In this work, for the first time the content of nitrite and nitrate ions in homogenized meat samples of baby foods was determined by a validated method based on ion chromatography (IC) coupled with conductivity detection. Recoveries of nitrate and nitrite ions in meat samples were not lower than 84 ± 6%. The detection limits of nitrate and nitrite were 0.08 mg L^−1^ and 0.13 mg L^−1^, respectively. Five commercial samples of homogenized meat, namely lamb, rabbit, chicken, veal, and beef, for infant feeding were investigated; while nitrite content was below the detection limit, nitrate ranged from 10.7 to 21.0 mg kg^−1^. The results indicated that nitrate contents were below the European (EU) fixed value of 200 mg kg^−1^, and an acceptable daily intake of 3.7 mg kg^−1^ was estimated.

## 1. Introduction

In the last few years, despite the world decline in birth rates, commercial baby food consumption increased due to the effortless alternative to home-made meals this food represents, especially for working parents in a nuclear family. For the commercial food preparation, more than 3000 additives are available as antioxidants and antimicrobial agents. Nitrate (NO_3_^−^) and nitrite (NO_2_^−^) salts are often preferred to inhibit the growth of bacterial spoilage in meat samples [1]. According to Commission Regulation (EU), No. 1129/2011, nitrates (sodium nitrate, E251; potassium nitrate, E252) and nitrites (potassium nitrite, E249; sodium nitrite, E250) are listed as permitted food additives. Beneficial effects of the addition of nitrates and nitrites to meat products are the improvement of quality characteristics as well as the microbiological safety. The nitrates and nitrites are mainly responsible for the development of the distinct flavor, the stability of the red color, as well as the protection against lipid oxidation in cured meat products. The nitrites show important bacteriostatic and bacteriocidal activity against several spoilage bacteria as well as foodborne pathogens found in meat products. The nitrites prevent the growth and toxin production by *Clostridium botulinum*. However, it has been demonstrated that high intakes of nitrate/nitrite could be detrimental to human health. Firstly, the nitrite ion can convert hemoglobin to methemoglobin, which causes methemoglobinemia (MHb), i.e., ‘blue baby syndrome’ [2]. This disease mainly affects children under one year of age, causing death by asphyxia. Secondly, nitrates are chemically unreactive but, through microbial reduction to nitrite ions, spur endogenous nitrozation reactions and can be precursors of nitrosamines, which are a cause of cancer in some animal species [3]. For these reasons, nitrates and nitrites taken from foods and water have been labelled by the International Agency for Research on Cancer (IARC) as probable human carcinogens [4,5,6]. Due to these potential adverse health effects, there are legal limits to control the maximum concentration of nitrate and nitrite allowed in different meat products (European commission Regulation N° 853/2004). Thus, it is imperative to determine the content of these ions in commercially available meat-based baby foods, by developing valid, reliable, and high-throughput analytical methods for the simultaneous determination of nitrate and nitrite. During the last 15 years, numerous methods have been reported in the literature for the separation and detection of nitrite and (or) nitrate [7] based on spectrophotometric [8,9], chemiluminescent [10], electrochemical [11], chromatographic [12], capillary electrophoretic [13], spectrofluorimetric [14], and electrochemiluminescent [15] techniques. However, spectrophotometric methods are subject to various interferences and lack of selectivity. As far as chromatographic methods, gas chromatography needs a derivatization reaction of both nitrite and nitrate, and liquid chromatography hides the risk of oxidation of nitrites, mainly when an acid medium is used [16,17]. Ion chromatography (IC) coupled with conductivity detection (CD) offers good reproducibility and high sensitivity and selectivity [18]. A literature search of nitrite and nitrate determination shows that IC-CD has been applied on several matrices, such as cheese [19], vegetables [20], juices fruit [21], and cured meats [22]. Besides, the European Committee for Standardization (EN 12014-4:2005) specified an IC-CD method for the determination of the nitrate and nitrite content of meat products, whose matrix is very different from the matrix of homogenized meat due to the presence of additives and fatty materials as well as the treatment process [23]. Up to now, nitrate and nitrite levels in infant foods were analyzed by using a spectrophotometric method [9] and few works focused on suppressed conductivity detection [24]. To our best knowledge, the IC-CD of nitrite and nitrate in commercial homogenized meats for infants and young children has been never reported. As homogenized meat is the best option for feeding babies between 6 and 12 months old, their exposure to nitrite and nitrate salts is greater in terms of body weight compared to older children. For this reason, the toxicological risk associated with the consumption of these foods containing nitrate and nitrite ions should be also estimated in term of daily intake in the light of the European Commission Regulation [25].

In this work, we report the applicability of an IC-CD method for the evaluation of nitrate and nitrite ions in five homogenized infant meats marketed in Italy: chicken, lamb, rabbit, beef, and veal.

## 2. Materials and Methods

### 2.1. Chemicals and Samples

All solutions were prepared using analytical grade reagents and deionized water. Eluent solution was composed by Na_2_CO_3_ (Carlo Erba ≥99.7%) and NaHCO_3_ (Carlo Erba 99.8%). Suppressor solution was prepared by H_2_SO_4_ (Carlo Erba, 95.0%). Nitrate and nitrite standard solutions were prepared by NaNO_3_ (purity >99.5%) and NaNO_2_ (purity 99.0%), purchased from Sigma–Aldrich (Milano, Italy). All the solutions used in this study were prepared by using ultrapure water supplied by Millipore Direct-Q UV unit (Bedford, MA, USA). Precautions were taken to minimize sample contamination. All sample containers and glassware were thoroughly cleaned with 3.65 g L^−1^ HCl solution and then finally with deionized water. The blank chromatograms with the double deionized water have shown no nitrite or nitrate peaks. Samples of baby meat foods were found in local supermarkets. Five different meat samples (chicken, lamb, rabbit, beef, and veal) were selected.

### 2.2. Instrument Setup

The 883 basic ion chromatography system (Metrom, Herisau, Switzerland) equipped with suppressed conductivity detector with Metrohm Suppressor Module (MSM; 50 mM H_2_SO_4_) was used for all analyses. Chemical suppression was accomplished by 4.91 g L^−1^ H_2_SO_4_, which was also used to adjust the baseline at an average value of 16.5 μS cm^−1^. Mobile phase was made up of Na_2_CO_3_ 190.78 mg L^−1^ and NaHCO_3_ 142.82 mg L^−1^ with a flow rate of 1.0 mL min^−1^. The injection was performed by using a 20 μL sample loop and the separation was performed at room temperature (20 °C) by using an isocratic gradient, consisting of Na_2_CO_3_/NaHCO_3_ buffer, and a Metrosep A SUPP-250 (250 × 4 mm, 9 μm particle size), which carrier material was an anion–exchange polymer of polyvinyl alcohol with quaternary ammonium groups. When the backpressure increases, double peaks or tailing effects occur, the retention times become shorter and the resolution deteriorates, the column regeneration is performed according Metrohm Manual (8.107.8040EN/2017-04-18) for Metrosep A Supp 5. Conductivity measuring system worked with a maximum pressure of 50 bar; a resolution of 0.0047 nS/cm; a noise <0.1 nS at 1 µS/cm and a measuring rate of 10 per second.

The MagIC net (version 2.4) software was used for monitoring system and data analysis. The ultrasonic bath was provided by Bandelin Sonorex RK 100 H (Bandelin electronic Berlin, Berlin, Germany).

### 2.3. Standard, Sample, and Solvent Preparation

Standard solutions were prepared at the bench from neat materials by using a metrologically valid procedure, including the evaluation of the purity of the neat reference material and the gravimetric preparation for both analyte and solvent, so that they may be considered to be purchased reference standards [26]. Nitrite and nitrate stock standard solutions (1000 mg L^−1^) in deionized water were prepared weekly, and then these standards were used to make mixed working standard for analysis. Four working standard solutions were prepared daily by diluting the mix standard with deionized water at final concentration ranged between 0 and 4 mg L^−1^, according to the experimental weights of corresponding salts: 0.456, 0.893, 1.78, and 3.42 for NaNO_2_ and 0.473, 0.945, 1.89, and 3.78 mg L^−1^ for NaNO_3_.

A total of 6 g of homogenized meat sample was diluted with boiling ultrapure water to obtain a final volume of 100 mL. Then, the samples were sonicated for 70 min at 50 °C. The mixture was then left to cool at room temperature and filtered through nylon filter membranes 0.22 μm Whatman (Maidstone, UK). Finally, 20 μL of samples were injected into the ion chromatographic system. All the extracted samples were taken and then stored at 4 °C for replicate analysis.

### 2.4. Validation Procedure

The validation process consisted of assessing the following parameters, according European Analytical Chemistry (Eurachem) guidelines [27,28]: linearity, limits of detection (LOD), limit of quantification (LOQ), precision, repeatability, trueness, and uncertainty.

Ten different aqueous standard solutions over the range 0–50 mg L^−1^ were analyzed to assess the linearity. Then, the calibration curves were calculated by analyzing four concentration levels (k = 4) prepared in the linear range of 0 to 4 mg L^−1^ for both nitrite and nitrate. Three replicate injections (*n* = 3) were performed at each level concentration, prepared in duplicate. Calibration curves (measurement response against a known concentration) were fitted by least-squares linear regression. To assess linearity, deviations of the mean calculated levels over three runs should be within 85–115% of nominal concentrations for the non-zero calibration standards [29]. The linearity was supported by evaluation of regression coefficient and statistical tests. The LOD and LOQ were estimated by regression parameters, respectively as $\frac{3.3s_{y/x}}{a}$ and $\frac{10s_{y/x}}{a}$, where the symbol $s_{y/x}$ is the residual deviation of the regression and a is the slope of the calibration curve.

The precision was assessed by calculating the percent relative standard deviation (%RSD) of repeatability (intra-day precision) and the intermediate precision (inter-day precision). For no significant matrix effect, the analysis of three replicates of each calibration standard in water matrix was performed on the same day for repeatability evaluation, and three non-consecutive days over three weeks for intermediate precision evaluation [29].

The trueness was determined through the recovery evaluation, by spiking real samples at 0.6 mg L^−1^ (10 mg kg^−1^), 1.2 mg L^−1^ (20 mg kg^−1^), and 2.4 mg L^−1^ (40 mg kg^−1^), which is in the middle of the linear working range of nitrite and nitrate. Then, the percentage recovery was calculated as average value of three independent replicates, by using this equation: %R = 100 × ((C_sample spk_ − C_sample_)/C_spk_) where C_sample spk_ is the analyte concentration in the spiked sample, C_sample_ is the analyte sample concentration and C_spk_ is the spiked analyte concentration. In addition, the study of the matrix effect is essential in order to assess the influence of the potential interferences, occurring in the real samples, on the analytical signal. Since no method blanks are available for this study, two simultaneous calibration curves were created and the corresponding slopes were compared: the first one was the calibration curve generated by analysis of standard solutions in water, the second one was prepared by adding to standards solutions the same portion of diluted homogenized meat extract [1].

The expanded uncertainty of experimental results was estimated as a combination of several contributions associated to the preparation of the concentration levels, the plotting of the calibration curves, the recovery and the limit of detection. The values of measurement uncertainty were expressed in terms of expanded uncertainty (U), which was determined by multiplying the combined standard uncertainty, u_c_, with a coverage factor k = 2; for normal distribution, this provides a level of confidence of approximately 95%.

## 3. Results and Discussion

### 3.1. Chromatographic Separation

Figure 1a (dotted line) shows a typical chromatogram of a mixed standard solution ([NO_2_^−^] 1.78 mg L^−1^ and [NO_3_^−^] 1.89 mg L^−1^). The repeatability was established by using a standard solution of nitrate and nitrite injected every 10–20 samples and calculating the mean of the observed retention times: 7.50 ± 0.03 min for nitrite and 10.77 ± 0.07 min for nitrate, according to a previously validated IC method [20]. The comparison between retention times of nitrite and nitrate in spiked meat (Figure 1a, continuous line) and for a standard mixture (Figure 1a, dotted line) showed the absence of interferences in the elution window. The chromatograms of a mixed standard solution, spiked sample (35 mg kg^−1^ of both nitrite and nitrate), and homogenized baby meat (chicken) are visible in plots (a) and (b) of Figure 1, respectively. Plot (b) reports an expanded view of the chromatographic profile of a homogenized chicken sample. At the same time, the content of nitrate ion was equal to 10.7 ± 0.2 mg kg^−1^ (see Table 1), and the level of nitrite was below the limit of detection (vide infra).

### 3.2. Method Validation

The validation of the analytical method was obtained from a diagnosis procedure of the capability of a method to determine the analytes under selected analytical conditions [27]. The quality parameters, as linearity, precision, trueness, and uncertainty, gained during the validation study are reported in Table 1.

The working ranges were 0.472–3.76 mg L^−1^ and 0.456–3.57 mg L^−1^ for nitrate and nitrite, respectively (Table 1). A good linearity of the analytical response was obtained under the IC-CD working condition (determination coefficient, r^2^ > 0.9993). The linearity was supported by statistical *t*-test to assess whether the determination coefficient is significant, by comparing the calculated value to t tabulated at 99.99% of confidence level.

%RSD values of intra-day and inter-day variation were less than 1.2%, according to the accepted reference value (15%) [30], thus confirming the suitability of this method for analyzing nitrite and nitrate in spiked meat samples. In detail, %RSD for repeatability was between 0.2% and 1.2% for nitrite and between 0.04% and 0.4% for nitrate. %RSD of intermediate precision was below 0.4% and 0.2% for nitrite and nitrate, respectively (Table 1). These results indicate that the IC-CD method has both good repeatability and inter-day precision.

The trueness was evaluated as the percentage recovery and matrix effect. According to the literature [31], the recovery study was performed using real samples fortified with three different concentrations of nitrite and nitrate: the recovery values ranged between 95% and 114% for nitrite and 84% and 103% for nitrate (Table 1). Typically, the highest recovery percentages were obtained with white homogenized meats. The comparison to literature data showed a good agreement with several matrices, thus suggesting complete extraction, minimal losses, good alignment between spiking and calibration solution and analytical system accuracy as well [22,32].

The matrix effect also affects the trueness due to possible interferences of some sample components with the analytes of interest. The presence and concentration of these interferent substances were obtained by comparison of the calibration curve generated by standard solutions in water with that obtained using homogenized meat samples, where nitrate already occurs (see Figure 2). As expected, without matrix effects the calibration curves are almost parallel [1,33]; the evaluated difference of slopes were 13.79% and 2.96% for nitrate and nitrite, plots (a) and (b) of Figure 2, respectively, of a homogenized chicken meat sample. Note that the curves of nitrite standard and matrix sample are almost coincident in Figure 2b, being nitrite ions absent in the investigated sample. Therefore, a matrix effect of less than 14% was observed when the calibration standards were prepared in ultrapure water. For the other meat samples under study, the matrix effect was lower than that observed for chicken sample in Figure 2.

Since the recovery value and matrix effect were within acceptance criteria [31], LOD and LOQ were estimated by regression parameters of standard solutions prepared in water. According to literature data [34], the LODs of nitrite and nitrate using a 20 μL loop were 0.13 and 0.08 mg L^−1^, respectively, and the LOQ values were 0.44 and 0.26 mg L^−1^ for nitrite and nitrate, respectively (Table 1).

Finally, the expanded uncertainty was calculated according to the Eurachem guidelines [28]. It ranged from 0.2 to 0.5 mg kg^−1^ for both nitrate and nitrite ions (Table 1).

### 3.3. Sample Analysis

Five commercial samples of homogenized meats for infant were analyzed: beef, rabbit, chicken, veal, and lamb. Although it can depend on the culture or cuisine, chicken and rabbit are generally classified as white meat, while red meat typically refers to beef, veal, and lamb. The biggest difference between the two is the nutritional content. The chromatographic profiles are illustrated in Figure 3. The quantitative content of nitrate and nitrite are summarized in Table 2. Nitrate occurred in all the five examined samples, in which the concentration ranged from 10.7 ± 0.2 to 21.0 ± 0.3 mg kg^−1^, and the mean value was 14.0 ± 0.2 mg kg^−1^. The lowest content occurred in the chicken sample; veal, lamb, and beef samples contained nitrate ranging between 12.6 ± 0.2 mg kg^−1^ and 13.3 ± 0.2 mg kg^−1^. The highest concentration (21.0 ± 0.3 mg kg^−1^) was found in the rabbit homogenized meat samples (Table 2). Previous studies reported nitrate values (mg/kg) ranged between 0.35 and 131.68, nitrite between 1.12 and 80.22. The highest mean values were found in baby foods of plant origin (45.5) for nitrate and in baby foods of mixed origin (12.48) for nitrite [9]. As expected for commercial foods, the level of nitrate ion was always lower than 200 mg kg^−1^ that is recommended by the EU regulation [25]. The absence of nitrite ion was ascertained in all meat samples; the peak signal of nitrite was below the LOD of the validated method.

### 3.4. Daily Intake

An acceptable daily intake (ADI) of 0–3.7 mg kg^−1^ body weight/day was established by the Scientific Committee of Food [35] and confirmed by the Joint Food and Agriculture Organization/World Health Organization (FAO/WHO) Expert Committee on Food additives [36]. Exposure to nitrate ions for each type of food depends on its concentration in food, and the amount of food consumed [37]. Nitrate exposure (mg nitrate kg^−1^ bodyweight day^−1^) is calculated as follows: (daily homogenized intake (mL) X mean concentration of nitrate in food (mg mL^−1^)/(body weight (kg)). The estimated daily intake (EDI) was calculated for infants from 3 to 12 months. The ADI of nitrite was set at 0.06 mg kg^−1^ bodyweight [38]. Table 3 lists the estimated nitrate intake values evaluated using consumption information provided by the DONALD study [39,40]. All the estimated values of nitrate intake are lower than the ADI values.

## 4. Conclusions

Health and toxic implications of nitrate and nitrite assumption led us to a thoughtful need to reconsider the analytical aspects of their quantitative determination. In this study, an ion chromatographic method was validated and demonstrated to be suitable for a simultaneous, accurate, and selective determination of nitrite and nitrate ions in homogenized meats for baby consumption. In detail, the obtained quality parameters, linearity, precision, trueness, and uncertainty, agreed with reference values and literature data. The method applicability established that the acceptable limit for the matrix effects in our validation procedure is less than 15%. According to EU regulation, in all examined meat samples, there was no-presence of nitrite whereas the nitrate content was relatively low. Data obtained were also in accordance with an acceptable daily intake value of 3.7 mg kg^−1^ of nitrates.

## Figures and Tables

**Figure 1 foods-09-01238-f001:**
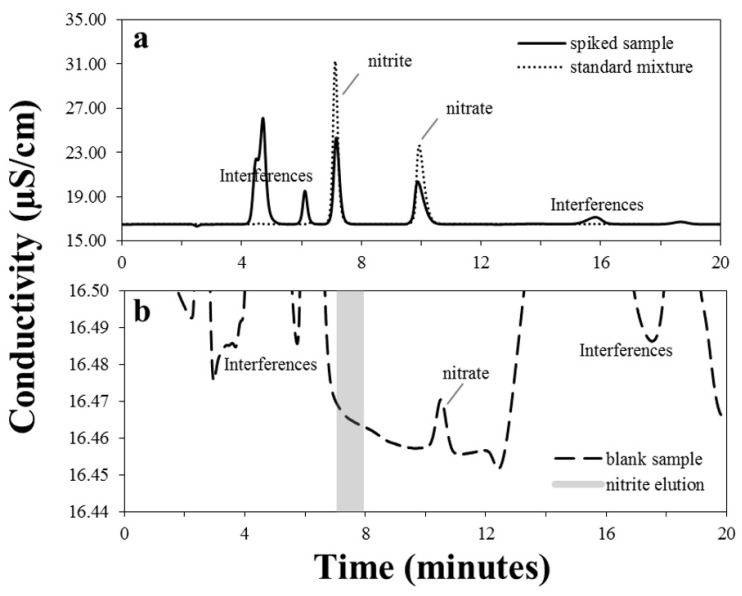
(**a**) Chromatographic profiles of a standard solution of nitrite 1.78 mg L^−1^ and nitrate 1.89 mg L^−1^ (dotted line) and a homogenized chicken sample, fortified with 35 mg kg^−1^ of both nitrite and nitrate (continuous line). (**b**) Ion Chromatography-Conductivity Detection (IC-CD) profile of naturally homogenized chicken (blank), where the grey shadow corresponds to the nitrite elution window.

**Figure 2 foods-09-01238-f002:**
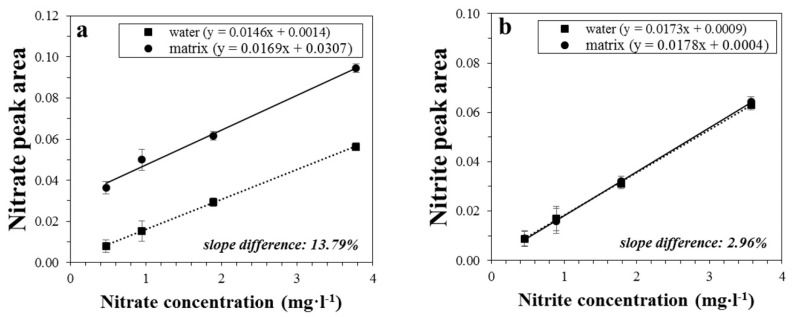
Calibration curves of nitrate (**a**) and nitrite (**b**) ions using standard solutions in water and a sample of homogenized chicken meat. Each point represents the mean of three replicated experiments (*n* = 3), and the vertical bars represent the standard deviation (SD).

**Figure 3 foods-09-01238-f003:**
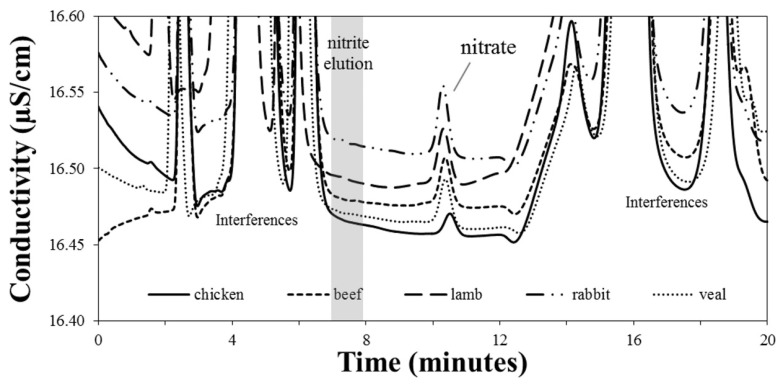
Comparison of five IC-CD chromatograms of samples prepared by homogenized lamb, rabbit, veal, beef and chicken meats. The expanded view shows each chromatographic profile.

**Table 1 foods-09-01238-t001:** Quality parameters of the IC-CD method validated for nitrite and nitrate determination: linearity (m, s_m_, b and s_b_ are the slope, the standard error of regression slope, the intercept and the standard error of regression intercept, respectively), precision expressed as percent relative standard deviation (%RSD), trueness evaluated as percentage recovery (%R), limit of detection (LOD), limit of quantification (LOQ), and uncertainty (U).

Quality Parameters		Analyte	
		Nitrite	Nitrate
Linearity	Linear range (mg L^−1^)	0.456–3.57	0.472–3.76
	Regression equation, y = (m ± s_m_)x + (b ± s_b_)	y = (17.3 ± 0.3)10^−3^x + (0.9 ± 0.7)10^−3^	y = (14.6 ± 0.1)10^−3^x + (1.4 ± 0.3)10^−3^
	Determination coefficient (R^2^)	0.9993	0.9998
	T_cal_ (T_0.05,4_ = 2.78, k = 6)	53.43	99.99
Precision, %RSD (concentration level, mg L^−1^)	Repeatability	0.2 (0.456)	0.04 (0.472)
		0.2 (0.893)	0.04 (0.945)
		0.6 (1.79)	0.1 (1.89)
		1.2 (3.57)	0.4 (3.78)
	Intermediate precision	0.1 (0.456)	0.2 (0.472)
		0.2 (0.893)	0.1 (0.945)
		0.4 (1.79)	0.1 (1.89)
		1.0 (3.57)	0.2 (3.78)
Trueness	%R (concentration, mg L^−1^)	95 ± 2% (0.6)	84 ± 6% (0.6)
		112 ± 11% (1.2)	87 ± 9% (1.2)
		114 ± 6% (2.4)	103 ± 2% (2.4)
	Matrix effect (slope difference %)	2.96	13.79
LOD (mg L^−1^)		0.13	0.08
LOQ (mg L^−1^)		0.44	0.26
Uncertainty, U (mg kg^−1^) (concentration level, mg L^−1^)		0.2 (0.6)	
		0.3 (1.2)	
		0.5 (2.4)	

**Table 2 foods-09-01238-t002:** The concentrations of nitrates and nitrites (mg kg^−1^), occurring in five homogenized meat samples, marketed in Italy. Values represent the means ± expanded uncertainty at 95% confidence level of *n* = 3 triplicate measurements (independently repeated experiments).

Sample	Nitrate (mg kg^−1^)	Nitrite (mg kg^−1^)
Beef	12.6 ± 0.2	<LOD
Rabbit	21.0 ± 0.3	<LOD
Chicken	10.7 ± 0.2	<LOD
Veal	12.6 ± 0.2	<LOD
Lamb	13.3 ± 0.2	<LOD

**Table 3 foods-09-01238-t003:** Estimated nitrate intake from homogenized meat selected for the study, calculated using consumption information provided by DONALD study and percentage of acceptable daily intake (ADI) for the different exposure scenarios using the mean, highest and lowest nitrate concentrations in all samples under study.

Consumption Scenarios			Estimation of Nitrate			% ADI		
Age (Months)	Body Weight (kg)	Mean Consumption of Baby Foods (g day^−1^)	Mean (14.04 mg kg^−1^)	Highest Value (21.0 mg kg^−1^)	Lowest Value (10.7 mg kg^−1^)	Mean (14.04 mg kg^−1^)	Highest Value (21.0 mg kg^−1^)	Lowest Value (10.7 mg kg^−1^)
3	5.80	67.0	0.16	0.24	0.12	4.38	6.56	3.34
6	7.50	195.0	0.36	0.55	0.28	9.87	14.76	7.52
9	8.60	234.0	0.38	0.57	0.29	10.32	15.44	7.87
12	9.40	208.0	0.31	0.46	0.24	8.40	12.56	6.40

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
