# Peer review of "Validation of an Analytical Method for Nitrite and Nitrate Determination in Meat Foods for Infants by Ion Chromatography with Conductivity Detection"

_foods, 2020, doi:10.3390/foods9091238_

Round 1

Reviewer 1 Report

The authors have not avoided minor editorial errors which do not reduce the quality of this work.

  1. Manuscript sections should be corrected: Materials and Methods should be the second section, after Introduction and before Results and Discussion, Conclusions
  2. In manuscript there are no information of the results of the nitrate and nitrite content in the baby meat samples were correlated with some other reference method
  3. In Table 1: there is no need to repeat the units in the second column in brackets (mg L-1), in the lines: Linearity range, Repeatability, Intermediate precision,% R and Uncertainty. Information on data units presented in parentheses is provided in the first column.
  4. In section 4.2. The setup instrument: did not provide all the parameters for the chromatographic analysis (i) was it an isocratic or a gradient separation? (ii) what temperature was maintained during the separation of the sample on the column? (iii) what were the conditions of the electrochemical detector?
  5. The information in lines 245-247 was repeated.
  6. Line 369: repeated "Statement on nitrites in meat products"
  7. Lines 375: should be "Lisbon"

Reviewer 2 Report

On lines 151/152 you state "the mean value wasn14.0±0.2" should read "the mean value was 14.0..."

In the introduction you describe the antimicrobial effects of nitrite, but on line 156 you state "Fortunately, the absence of nitrite ion was ascertained in all meat samples;the peak signal of nitrite was below the LOD of the validated method." It is clear that excessive consumption of nitrite is detrimental, but the absence of it can lead to the growth of undesirable bacteria. Perhaps remove the word Fortunately.

Did each of the food products have nitrite added to them? can you state this in the methods. If not, can you describe how naturally occurring nitrate may be converted to nitrite.

Reviewer 3 Report

The manuscript entitled "Validation of an analytical method for nitrite and nitrate determination in meat foods for infant by ion chromatography with conductivity detection" aims to describe the validation and aplication of an analytical method to the analysis of nitrite and nitrate in baby food samples prepared with homogenized meat by ion chromatography with conductivity detection. 

The work is interesting, although the novelty only resides in its aplication, as the chromatographic method seems to not have been optimized. Additionally, some issues in the structure of the paper should be addressed.

According to the journal guidelines, the structure of the article section should be: Introduction, Materials and Methods, Results and Discussion, Conclusions. Authors need to change the order of the sections, as they have provided the materials and methods section as the last one. Therefore, they should revise and restructure the article according to the journal guidelines.

The tables provided should be improved to increase the reader comprehension. The size of the font letter should be reduce for some headings, so they are not cut hindering the reading. E.g. in Table 1 the heading of precision is cut, as well as the headings in Table 3.

This reviewer has some concerns about the procedure followed for the validation of the method proposed. As authors stated in their introduction, nitrites and nitrates are additives well regulated, for which maximum levels have been stablished (CE Regulation 57 N° 853/2004). Therefore, usually, when compounds in food are legislated, some guidelines for the development and validation of analytical methods are also provided or indicated. Nevertheless, the validation procedure carried out by the authors does not follow any official regulation for these compounds in food products. Indeed, authors only indicate that validation parameters were evaluated according to reference 26. However, when they explain how the parameters were calculated in the materials and methods section, they referrence some of their previous works [40, 42, 43], what seems as inappropriate self-citations. Therefore, this reviewer belives that official guidelines for validation of analytical methods should be referrenced in these sections instead of the works referrenced by the authors.

The section 4.2 Intrument setup would be more appropriate to be named chromatographic analysis. Authors should provide details about the chromatographic separation, such as the mobile phase (its composition and type of gradient) the analysis run time, retention time, the column regeneration, etc. Some of this information is provided in the next section 4.3 (lines 223 to 226), which would be more appropriate to be moved to section 4.2. 

Which was the sample used for the evaluation of the matrix effects? was this matrix effect evaluated with all the samples analysed? What criteria was followed to select the validation levels for precision? and for trueness? why were not the same for both parameters? All this should be explained by the authors in the manuscript. 

Please, use "blank sample" instead of "white sample". This should be corrected in the text. Moreover, authors explain in section 4.4 "Since no method blank are available for this study [...]", however, in section 2.2 they stated "Typically, the highest recovery percentages were obtained with white homogenized meats". Therefore, if no blank samples were available, how the authors used them for the evaluation of the trueness parameter? this should be explained. 

Minor comments:

  • Abbreviation should be defined at first mention, and then used consistently thereafter. E.g. authors talk all thoughtout the manuscript about nitrites and nitrates, but in lines 85-86 they refer to them with their chemical formula. 
  • Although haemoglobin and methaemoglobin are correct terms, better use hemoglobin and methemoglobin as these terms are more frequent. Indeed, the issue methemoglobinemia mention after it is more consistent with the latter terminology. 
  • check line 151 "wasn", correct this

Round 2

Reviewer 3 Report

Thank you for the revised version of the manuscript. Authors have addressed many of the changes suggested to improve the manuscript. Nevertheless, I still have some minor comments:

  1. Please, revised again the reference list. E.g. references 29 and 31 do not appear on the text but they are listed in the reference list. 
  2. The explanation of the "white samples" provided by the authors is fine "... white sample does not mean “blank sample”. For “white sample” we mean the white meat, as reported at L207-208. Although it can depend on the culture or cuisine, white meat is generally classified as poultry (chicken and turkey), while red meat typically refers to beef, pork, and lamb. The biggest difference between the two is nutritional content" However, this explanation should be included in the manuscript to avoid misunderstandings. Please, specified it on the manuscript.

Author Response

Response to Reviewer 3 comments

POINT 1. Please, revised again the reference list. E.g. references 29 and 31 do not appear on the text but they are listed in the reference list. 

RESPONSE 1. We are so sorry for this oversight but Mendeley did not work! We amended manually the references' list.

POINT 2. The explanation of the "white samples" provided by the authors is fine "... white sample does not mean “blank sample”. For “white sample” we mean the white meat, as reported at L207-208. Although it can depend on the culture or cuisine, white meat is generally classified as poultry (chicken and turkey), while red meat typically refers to beef, pork, and lamb. The biggest difference between the two is nutritional content" However, this explanation should be included in the manuscript to avoid misunderstandings. Please, specified it on the manuscript.

RESPONSE 2. At L238-240, we added “Although it can depend on the culture or cuisine, chicken and rabbit are generally classified as white meat, while red meat typically refers to beef, veal, and lamb. The biggest difference between the two is the nutritional content.”

This manuscript is a resubmission of an earlier submission. The following is a list of the peer review reports and author responses from that submission.